# Mutational Analysis of EGFR Mutations in Non-Small Cell Lung Carcinoma—An Indian Perspective of 212 Patients

**DOI:** 10.3390/ijerph20010758

**Published:** 2022-12-31

**Authors:** Amrit Kaur Kaler, Khushi Patel, Harshali Patil, Yash Tiwarekar, Bijal Kulkarni, Meenal Hastak, Nivetha Athikari, Samrudhi Rane, Ankita Nikam, Smita Umarji, Imran Shaikh, Sandeep Goyle, Rajesh Mistry

**Affiliations:** 1Department of Molecular Pathology and Genomics, Kokilaben Dhirubhai Ambani Hospital, Mumbai 400053, India; 2Department of Pathology, Kokilaben Dhirubhai Ambani Hospital, Mumbai 400053, India; 3Department of Oncology, Kokilaben Dhirubhai Ambani Hospital, Mumbai 400053, India

**Keywords:** non-small cell lung carcinoma, EGFR, pathogenicity, exons, squamous cell carcinomas, adenosquamous carcinomas, sarcomatoid carcinomas, large cell carcinoma, metastasis

## Abstract

Lung cancer is the world’s leading cause of cancer-related deaths. Epidermal growth factor receptor (EGFR) is one of the critical oncogenes and plays a significant role in tumor proliferation and metastasis. Patients with sensitizing mutations in the EGFR gene have better clinical outcomes when treated with tyrosine kinase inhibitors (TKI). This study expands our knowledge of the spectrum of EGFR mutations among lung cancer patients in the Indian scenario. This is a retrospective descriptive study of all newly diagnosed patients with lung cancer in tertiary care hospital in India. All the samples were subjected to real-time PCR (q-PCR) analysis and confirmation of rare novel mutations was done using Sanger sequencing. Clinicopathological characteristics, mutational EGFR status, and location on the exon and metastatic sites were evaluated. An analysis of total 212 samples showed mutations in 38.67% of cases. Among these, five (5.9%) samples had mutations in exon 18, 41 (48.8%) samples had mutations in exon 19, 12 (14.28%) samples had mutations in exon 20, and 26 (30.95%) samples had mutations in exon 21. Eleven (13.41%) were found to be uncommon EGFR mutations. Additionally, six (21.4%) samples that had EGFR mutations were also positive for brain metastasis. Future testing on bigger panels will help to characterize the incidence of genetic mutations and to determine the appropriate targeted treatment choices for NSCLC patients.

## 1. Introduction

Lung cancer is the second most diagnosed cancer (11.4%) and the most prevalent cause of cancer-related mortality, accounting for 18% of all cancer-related deaths [1]. The high pervasiveness of lung cancer has contributed to new discoveries in various fields of genetics, new treatment trials, public health, and improved drug discoveries [2].

Primary lung cancer is divided into two major groups, small cell lung cancer (SCLC) and non-small cell lung cancer (NSCLC). The more common of these two is the NSCLC, which constitutes approximately 80% to 85% of all lung cancers, while only 15% of cases are diagnosed as SCLC. The majority of lung cancer patients present with metastatic or advanced disease at the time of diagnosis [3,4].

Smoking is considered to be the leading cause of lung cancer, accounting for 85% of all cases, along with other risk factors such as passive smoking, asbestos, radon, and environmental factors. [5]

One of the most common genetic alterations that were observed in NSCLCs is EGFR mutations, accounting for 33.07% of cases. It is more common in East Asians, non-smokers, and females [6]. Activating mutations of EGFR are present in the segment comprising exons 18 to 21, which codes for the EGFR protein tyrosine kinase domain [7]. The drug-sensitive mutations are as follows: point mutations in exon 18 (p.G719A, p.G719C, and p.G719S), point mutations in exon 21 (p.L858R and p.L861Q), and in-frame deletions in exon 19. The drug-resistant mutations are present in exon 20 (p.T790M) [8]. The most common mutations observed in patients with advanced NSCLC are exon 19 deletions and the L858R point mutation in exon 21 [9,10].

We present a study of 212 cases with a spectrum of mutations in the EGFR gene based on clinicopathological studies, molecular characterization, location on exon, and their association with the site of metastasis.

## 2. Materials and Methods

### 2.1. Study Design, Population Selection and Data Collection

A retrospective cross-sectional study was carried out of all the newly diagnosed cases of NSCLC (*n* = 212) from primary or metastatic sites in a tertiary care hospital in West India from January 2019 to March 2022, following the workflow given in Figure 1. The study parameters included age, gender, histopathological characteristics, type of EGFR mutation based on the location of exon, and association with the site of metastasis. Ethical approval for the study was obtained from the Institutional Ethics Committee-Academics (Registration no. EC/NEW/INST/2022/MH/0046). Project identification code for our paper is KDAH/PUB/2022/37 and has received its approval on 7 September 2022.

The inclusion criteria chosen were as follows: 1. Patients must be confirmed morphologically and classified according to WHO as lung cancer. 2. All patients must be above 18 years. 3. The chosen EGFR exons 18, 19, 20, and 21 will only be included in the study. The exclusion criteria was histopathologically diagnosed cases of small cell carcinoma.

### 2.2. Tissue Preparation and DNA Extraction

Tumor EGFR mutation status was determined from biopsy samples of lung cancer-diagnosed patients by analyzing DNA extracted from formalin-fixed paraffin-embedded (FFPE) using QIAamp DNA FFPE Tissue Kit, which allows rapid and efficient purification of genomic DNA. Tumor cells were collected in ribbons using a microtome and placed in microcentrifuge tubes. Xylene (1 mL) was added and centrifuged, the supernatant was discarded, and 1 mL ethanol (96–100%) was added. After incubation, the pellet was resuspended in 180 uL of buffer ATL, and 20 uL of Proteinase K. After incubation and centrifugation, 200 uL of buffer AL and 200 uL of ethanol (96–100%) were added. The entire lysate was transferred to the QIAamp column, and 500 uL of buffer AW1 was added and centrifuged, followed by the addition of AW2 buffer. The collection tube was discarded and centrifuged at full speed (20,000× *g*; 14,000 rpm) to pellet the debris.

### 2.3. Detection of EGFR Mutations

Histopathological analysis ensured that samples were adequate and appropriate for use. All samples were amplified using eight different mixes: EGFR G719x, EGFR T790M, EGFR S768I, EGFR ex20ins, EGFR L858R, EGFR L861Q, EGFR ex19del, and EGFR ctrl, and tested using an amplification refractory mutation system (ARMS)-based EGFR mutation detection kit. The ARMS kit is able to detect 30 mutations: exon 18 (G719A, G719S, and G719C; the kit is unable to distinguish between these subtypes, which are referred to as G719X), 19 deletions in exon 19, two mutations in exon 20 (S768I, T790M), four insertions in exon 20, and two mutations in exon 21 (L858R, L861Q). The EGFR mutation status of each patient’s tumor was assessed from the individual status of all EGFR mutation types and recorded as one of the following: positive (mutation detected by the assay) or negative (no mutation detected by the assay).

### 2.4. EGFR Mutation Screening

All the samples were subjected to real-time PCR (q-PCR) analysis to determine the presence of mutations in the EGFR gene, followed by Sanger sequencing to detect certain rare mutations obtained from tumor cells in some cases.

**Real-time PCR**—Mutational analyses of EGFR were performed using real-time polymerase chain reaction (RT-PCR) fragments amplified from genomic DNA. The Easy^®^ EGFR kit was used, and each sample was amplified using different reagent mixes. Amp-Mix (15 µL) was pipetted into wells. Detection was performed in ABI 7500 Fast: MICROAMP FAST OPTICAL 96 WELL RNX PLATE code 4346907; OPTICAL ADHESIVE COVERS code 4360954. However, it did not identify exact sequence changes.

**Sanger-sequencing**—This was performed to confirm 11 rare possible point mutations with specific amino acid changes reported in the literature. Two primers, a forward and reverse primer, were used for each exon. All the primers were reconstituted using molecular grade RNAse and DNAse-free water. For each exon, 18 uL of reagents-HS Buffer, Q Solution, dNTPs, primers, water, and HS Taq were added to each tube. DNA was added to the positive control, samples, and negative control. PCR products were electrophoresed on 2% agarose gels along with 3 µL of 100 bp DNA Ladder. EXOSAP (4 µL) was added, and bidirectional sequencing was done using the ABI 3500DX sequencer.

### 2.5. Statistical Analysis

Statistical significance was defined as *p*-value less than 0.05. The results were analyzed using SPSS software(Version 25,2007, IB, Corporation, Armonk, New York, United States).

All continuous variables were presented as mean ± standard deviation (SD), and categorical variables were presented as numbers (%). *p*-value < 0.05 (two-tailed) was considered statistically significant. All statistical analyses were calculated using SPSS version 24.0 (IBM Corporation, Armonk, NY, USA).

Descriptive statistics (frequency, median, and mean) were used to calculate the demographic and clinical characteristics. Categorical comparisons were calculated by the chi-square test or by Fisher’s exact test. Continuous variables were compared using the *t*-test.

Statistical significance was set at *p* < 0.05 for all analyses. All analyses were performed using the software IBM SPSS (Version 25,2007, IB, Corporation, Armonk, New York, United States).

## 3. Results

### 3.1. Demographic and Histological Characteristics of Patient Samples

A total of 212 newly diagnosed lung cancer patient biopsy and resection specimens from primary and metastatic sites were tested to identify the type and number of EGFR mutations along with their molecular and clinical characteristics. The incidence of EGFR mutations was found to be 38.6% (82 cases). The mean age of the total population was 62.05 years, with 40.65% patients below 60 years and 59.34% above 60 years. EGFR mutation prevalence was higher in females (*n* = 45; 54.88%) and in older adults aged > 60 years (*n* = 49; 59.75%), as shown in Table 1. Considering EGFR mutation status, 45 of 82 (54.9%) patients were females, and 37 (45.12%) were males, which showed a significant *p* value < 0.05. Males above 60 years of age showed the highest prevalence of EGFR mutations (*n* = 25; 67.5%) but with no significant *p*-value.

With respect to histopathological subtypes, 173 (82.4%) were adenocarcinoma, 17 (7.9%) were squamous cell carcinoma, 14 (6.54%) were adenosquamous carcinoma, three (1.4%) were sarcomatoid carcinoma, and three (1.4%) were undifferentiated carcinoma. Adenocarcinoma samples constituted the largest percentage (*n* = 71; 86.58%) of EGFR mutated samples, but the *p*-value was not found to be significant (*p* = 0.379), as seen in Table 1.

### 3.2. EGFR Mutation Screening and Detection

EGFR mutations were detected in 82 (38.67%) patients, of which five (5.9%) were in exon 18, 41 (48.8%) in exon 19, 12 (14.28%) in exon 20, and 26 (30.95%) in exon 21, as seen in Figure 2. There was one (1.22%) sample that had three mutations and six (7.32%) samples that had two mutations when tested. Exon 19 in-frame deletions (c.2235_2249del, p.Glu746_Ala750del; c.2236_2252del, p.E746_T751del; c.2240_2254del p.Leu747_Thr751del) at exon 19 (*n* = 38; 45.2%) and the p.L858R substitution mutation in exon 21 (*n* = 23; 27.38%) were detected as the most common mutations among all positive samples. Tumor samples of two patients (1.4%) had T790M mutation in exon 20.

Table 2 provides the exon location of EGFR mutations according to different lung cancer types. Only adenocarcinoma samples showed EGFR mutations in all four exons tested, i.e., 18, 19, 20, and 21. Squamous cell carcinoma and adenosquamous carcinoma had mutations in three of the four exons, 18, 19, 21, and 18, 20, 21, respectively, carcinosarcoma in exon 21, undifferentiated carcinoma in exon 19, and sarcomatoid carcinoma in exon 18. Two of three adenosquamous carcinoma patients showed double mutations.

### 3.3. Molecular Characterization of EGFR Mutations

Table 3 provides details about the location and type of each mutation, its corresponding amino acid change, along with its metastatic status. As per ACMG classification, each mutation was classified into three categories—pathogenic, likely pathogenic, and benign, which helps to predict the behavior of the mutation. EGFR mutations were classified according to the above classification, as inferred in Figure 3 [11]. Exon 19 had the highest number of pathogenic mutations, contributing 41 (45%) out of a total of 82 identified EGFR mutations. Out of the 41 mutations, 17 (66.7%) were deletion, three (22.2%) were deletion. One (11.1%) substitution mutation was reported as benign. Three (22.2%) of the deletion mutations of exon 19 showed metastasis to the brain in a total of six patients.

Exon 20 had 12 (25%) mutations, of which eight (80%) were substitutions, and four (20%) were insertion mutations. Five of the mutations were pathogenic and metastatic, whereas one was found to be pathogenic but not metastatic and of the remaining six and four were benign and non-metastatic, respectively, and two were insertions. Exon 18 had six (20%) mutations, four of which were substitutions, three (75%) were pathogenic, and one (25%) was likely pathogenic; none of the mutations were metastatic. The remaining one was a novel variant of exon 18 and the other was not sequenced to understand its cDNA and amino acid change. Exon 21 had the second highest number of mutations, which was 26 (10%), of which 25 were substitutions and pathogenic, and L858R could possibly be metastatic or not. However, L861Q is metastatic.

#### Uncommon Mutations of EGFR

Out of all uncommon EGFR mutations, 11 (13.41%) were found to be present to a lesser extent in exons 18, 19, 20, and 21 (*n* = 2, *n* = 4, *n* = 3, *n* = 2), respectively, which included G719X (18.18%) in exon 18; S768I in exon 20 and exon 20 insertions; and L861Q in exon 21.

### 3.4. Metastatic Location and Features of EGFR Mutations

Considering metastasis status, lymph node metastasis was the most common (*n* = 35; 44.9%) among all samples, followed by pleural effusion (*n* = 9; 11.5%) and brain metastasis (*n* = 8; 10.3%). Other organs to which the cancer had metastasized were bone (*n* = 5; 6.4%) and spinal cord (*n* = 2; 2.6%), followed by contralateral lung (*n* = 1; 1.26%) and pericardial fluid (*n* = 1; 1.26%). Fifteen patients (19.2%) had multiple metastases, out of which five (17.9%) patients were EGFR positive. Among the EGFR mutation positive samples, 12 (42.9%) showed lymph node metastasis, followed by the brain (*n* = 6; 21.4%), multiple sites (*n* = 5,17.95%), two (7.1%) each showed pleural fluid and bone metastases, and one (3.6%) sample showed metastasis to the femur. Metastasis was found to be higher in brain among EGFR positive cases (21.4%) as compared to negative cases (4.0%). Classification of metastatic samples based on EGFR mutation is summarized in Table 4.

Table 5 provides the list of EGFR mutations present in brain metastatic samples. Out of a total of six EGFR-positive samples that showed brain metastasis, three (50%) samples had mutations in exon 19 and three (50%) samples had mutations in exon 21. Among the three exon 19 mutations present in brain metastasis samples, two (66.7%) were deletions and one (33.3%) was a deletion and insertion mutation. The other three samples showed the same mutation in exon 21, which was an L858R substitution mutation, contributing 50% of all mutations in brain metastasis samples.

## 4. Discussion

The EGFR gene, also known as ERBB, HER1, mENA, and proto-oncogene c-rbB-1, is a protein-coding gene. It codes for the epidermal growth factor receptor, which is a transmembrane receptor protein. Mutations in the EGFR gene lead to the production of a protein that is constitutively activated, resulting in constant activation [12,13]. Girard (2018) reported a progression free survival of 10–14 months among EGFR responders started on tyrosine kinase inhibitors (TKI) [14].

In the present study, 38.7% of the patients harbored EGFR mutations. Noronha, V., (2013) reported an EGFR mutation rate of 35% among NSCLC in the Indian population [15] A large-scale Asian study on adenocarcinoma patients found the overall EGFR mutation rate to be 51.4%, while it was reported to be 30% in the East Asia population,. Considering the worldwide incidence of EGFR mutations in NSCLC patients, the positivity rate is 10 to 15% in North America and Europe [16], and 19% in African–American patients [17].

The EGFR mutation rate of female patients in our study is 54.88%, which is comparable to Asian population with a mutation frequency of 61.1%. Noranha also reported a similar finding stating a gender predilection with positivity rate of 69% in females. The present study showed the high incidence of EGFR mutations in males over 60 years of age (67.5%) without a significant *p* value. Chougule A et al. (2013) also reported a marginally high mutation rate in males above 60 years as compared to those aged less than 60 years [18].

With respect to histomorphology, our results show that the rate of EGFR mutations is highest (82.4%) in adenocarcinoma, followed by squamous cell carcinoma (7.9%) and other types of lung carcinomas (10.27%). This histological prevalence is similar to other studies [19,20], which have reported high EGFR mutation rates in patients with adenocarcinoma. Noronha et. al. reported 39 lung cancer patients with EGFR mutations, out of which 38 were adenocarcinoma patients, and one patient had squamous cell carcinoma.

The type of EGFR mutations with respect to the location has been shown to be an important factor to understand the prognosis of TKI inhibitors [21]. In our study, the mutations were distributed primarily in exons 19 (48.8%) and 21 (30.95%), constituting a total of 79.76% of all mutations. Exon 18 showed five (5.9%) mutations, and exon 20 had 12 (14.28%) mutations. Yoon et. al. also reported similar results; a study conducted on NSCLC patients of a Caucasian population reported the prevalence of EGFR mutations in exons 18, 19, 20, and 21 to be 7.41%, 46.3%, 9.26%, and 37.04%, respectively. A study conducted by Chougule A et al. (2013) on the Indian population found the EGFR mutation rate to be highest in exon 19 (50%), followed by exon 21 (42%), exon 18 (7%), and exon 20 (3%). In-frame deletions in exon 19 are the most frequently reported EGFR mutations in lung cancer patients present at a rate of around 46.3% to 76.2%, followed by L858R missense mutation in exon 21 present at a rate of 26% to 38% as reported by Noronha et. al. 

Out of all uncommon EGFR mutations, 11 (13.41%) were found to be present to a lesser extent in exons 18, 19, 20 and 21 (*n* = 2, *n* = 4, *n* = 3, *n* = 2) respectively, which included G719X (18.18%) in exon 18; exon 19 rare deletions (36.36%), and S768I in exon 20 and exon 20 insertions (27.3%); and L861Q in exon 21 (18.18%). This was concordant with a study done by Hai-Yan Tu stating the frequency of uncommon mutations as 11.9% [22]. Rare EGFR mutations have been reported to be present at a rate of 12% in adenocarcinoma patients, with G719X having a prevalence of 13.78%, S768I/V having a prevalence of 6.39%, exon 20 insertions having a prevalence of 16.85%, and L861X having a prevalence of 9.88% [23]. These results are higher than a study by Zhou F et al. that reported the rates of rare mutations; for instance, G719X in exon 18 is present at a rate of 7.9%, and L861Q in exon 21 is present at a rate of 9.8%. Our study also detected the presence of T790M mutation in exon 20, which has been shown to confer resistance to treatment with third generation TKIs [24,25].

This study also identified likely correlations between the type of mutations and the metastatic status in patients with EGFR mutations. Lymph node metastasis was highest, followed by brain metastasis, where the prognostic outcome was poor. Out of 20.69% patients with brain metastasis, almost all resulted in death after a certain period. This was common in patients having L858R exon 21 mutation, and they were treated with chemotherapy or chemo-immunotherapy or radiation therapy. A possible approach to prevent brain metastasis could be timely diagnosis with rapid techniques like real-time PCR and targeted treatment with EGFR-TKI [26]. Studies have shown a strong correlation between EGFR mutations and brain metastasis [27,28]. A meta-analysis conducted using 22 studies (*n* = 8152) showed a high association between EGFR mutations and brain metastasis with an odds ratio of 1.99 [29]. Shin et al., also showed that while EGFR mutation status was correlated with brain metastasis, there was no association between EGFR mutations and extracranial metastases. The L858R mutation in exon 21 has been shown to be strongly associated with brain metastasis similar to our study, where 50% of the brain metastasis samples harbored this mutation [30], whereas the presence of exon 19 deletions has been reported to result in characteristic radiographic features in brain metastasis samples [31]. In contrast, a recent meta-analysis report by Beak et. al. has reported a stronger propensity of patients with exon 19 deletions to have brain metastasis as opposed to exon 21 mutations, possible owing to greater acceptance of therapy with TKIs in exon 19 mutation lung cancer cells.

## 5. Conclusions

EGFR mutations are an important part of the diagnostic modality towards treating NSCLC. We found an incidence of 38.7% of EGFR mutations in our study population. There was a strong association of EGFR mutations with adenocarcinoma, and older age population (>60 years), which showed a significant *p*-value of 0.005. Exon 19 and exon 21 EGFR positive were the most common mutations noted in the study. For EGFR-positive lung adenocarcinoma patients, exon 21 L858R mutation showed a maximum propensity towards metastasis, hence poor prognosis. A possible approach of screening all the newly diagnosed lung carcinoma patients for the established biomarkers could help in better treatment planning. Moreover, large-scale prospective studies will be needed to understand the link between the pathogenicity of the rare mutations and therapeutic strategies like EGFR-TKI, to treat some resistant and uncommon EGFR mutations in NSCLC.

## Figures and Tables

**Figure 1 ijerph-20-00758-f001:**
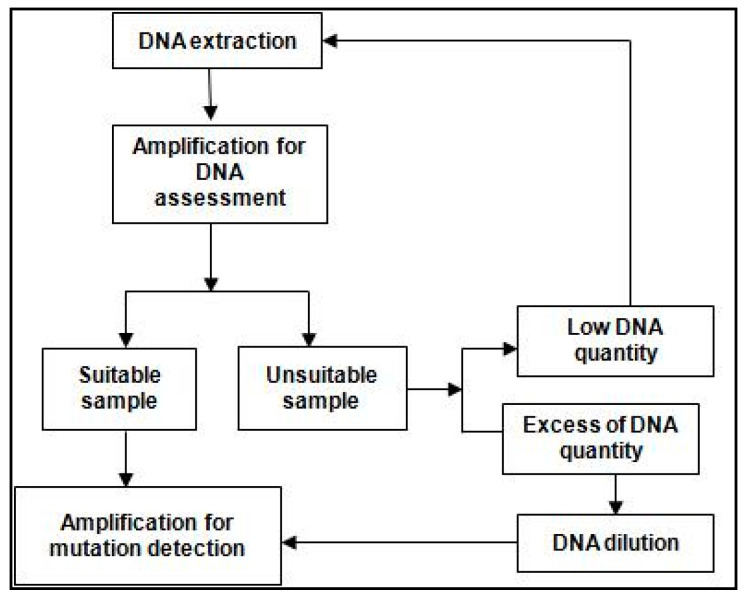
Workflow for screening of patient samples for EGFR mutations.

**Figure 2 ijerph-20-00758-f002:**
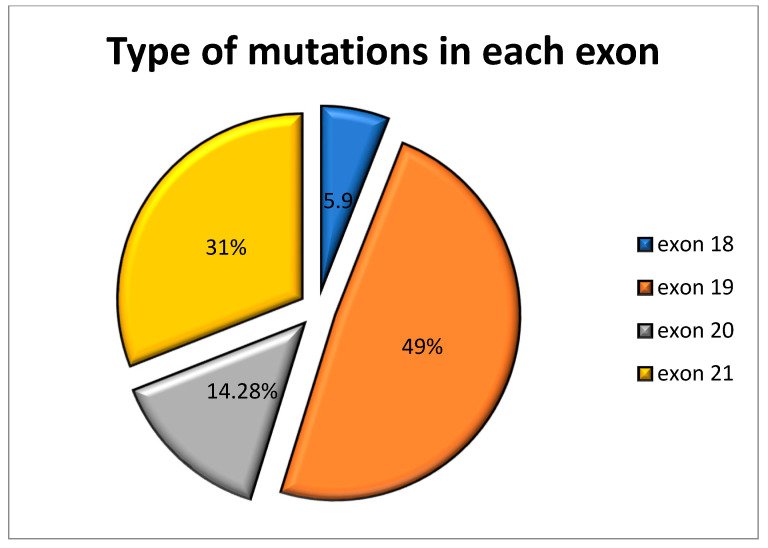
Percent representation of mutations present in EGFR gene exons 18, 19, 20, and 21.

**Figure 3 ijerph-20-00758-f003:**
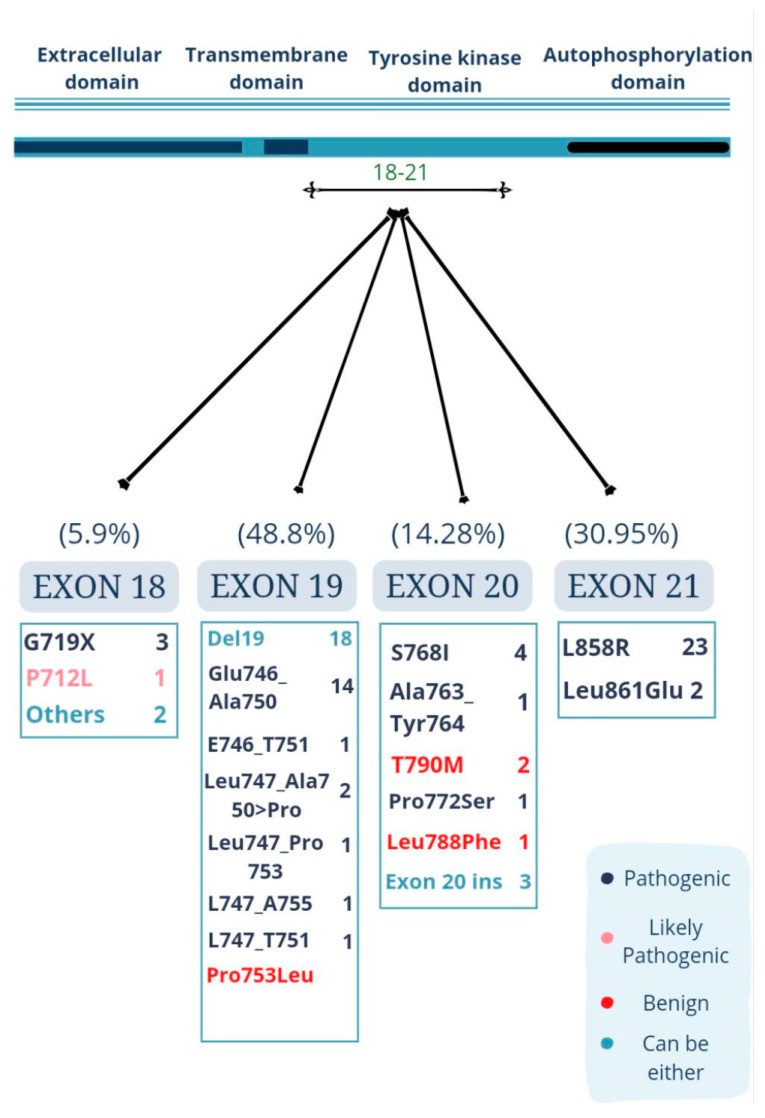
Pathogenicity highlights for EGFR mutations and categories.

**Table 1 ijerph-20-00758-t001:** Demographic and histological characteristics of total, EGFR mutated, and non-EGFR mutated patient samples.

	Global Population (*n* = 212)	EGFR Mutated Patients (*n* = 82; 38.67%)	EGFR Non-Mutated Patients (*n* = 130; 61.32%)	*p* Value
Demographic characteristics				
Male	121 (57%)	37 (45.12%)	84 (64.6%)	0.007
Female	91 (43%)	45 (54.88%)	46 (35.38%)	0.007
Male (≤60)	42 (34.71%)	12 (32.4%)	30 (35.7%)	0.726
Male (>60)	79 (65.29%)	25 (67.5%)	54 (64.3%)	0.726
Female (≤60)	41 (45%)	20 (44.44%)	21 (45.6%)	0.907
Female (>60)	50 (54.94%)	25 (55.5%)	25 (54.3%)	0.907
Histological characteristics				
Adenocarcinoma	173 (82.4%)	71 (86.58%)	102 (77.7%)	0.379
Squamous cell carcinoma	17 (7.9%)	5 (6.09%)	12 (9.09%)	0.379
Adenosquamous carcinoma	14 (6.54%)	3 (3.65%)	11 (8.33%)	0.379
Sarcomatoid carcinoma	3 (1.4%)	2 (2.43%)	1 (0.75%)	0.379
Large cell carcinoma	3 (1.4%)	1 (1.21%)	2 (1.51%)	0.379

**Table 2 ijerph-20-00758-t002:** Histological classification of lung cancer types based on the location of EGFR mutations.

Histology Type	Number of Samples	Exon
Adenocarcinoma	71	18, 19, 20, 21
Squamous cell carcinoma	5	18, 19, 21
Adenosquamous carcinoma	3	18, 20, 21
Large cell carcinoma	1	19
Sarcomatoid carcinoma	3	18, 21

**Table 3 ijerph-20-00758-t003:** Types of most common EGFR mutations, exon locations, and metastatic sites.

Exon No.	Number of Mutations	cDNA Change	Amino Acid Change	Type of Mutation	Metastasis
18	1	c.2156G>C	Gly719Ala	Substitution	No
18	1	c.2155G>A	Gly719Ser	Substitution	No
18	1	c.2134T>C	Phe712Leu	Substitution	No
18	1	c.2155G>T	Gly719Cys	Substitution	No
19	7	c.2235_2249del	Glu746_Ala750	Deletion	No
19	7	c.2236_2250del	Glu746_Ala750	Deletion	No
19	1	c.2236_2252del	E746_T751	Deletion	No
19	2	c.2239_2248delinsC	Leu747_Ala750>Pro	Deletion and Insertion	No
19	1	c.2240_2257del	Leu747_Pro753	Deletion	Yes
19	1	c.2240_2267delinsGCCAA	Leu747_Ala755	Deletion and Insertion	Yes
19	1	c.2258 C>T	Pro753Leu	Substitution	No
19	1	c.2240_2254del	Leu747_Thr751	Deletion	Yes
20	4	c.2303G>T	S768I	Substitution	Yes
20	1	c.2290_2291insTCCGGGAAGCCT	Ala763_Tyr764insPheArgGluAla	Substitution	Yes
20	1	c.2314C>T	Pro772Ser	Insertion	No
20	1	c.2362 C>T	Leu788Phe	Substitution	No
20	2	c.2369C>T	T790M	Substitution	No
21	23	c.2573T>G	L858R	Substitution	No/Yes
21	2	c.2582T>A	L861Q	Substitution	Yes

**Table 4 ijerph-20-00758-t004:** Classification of metastatic samples based on EGFR mutation status and location of metastasis (F—Frequency; P—Percentage).

	Frequency of EGFR Negative Mutation (*n* = 50)		Frequency of EGFR Positive Mutation (*n* = 28)		Total (*n* = 78)	
	Total	Percentage (%)	Total	Percentage (%)	Total	Percentage (%)
Lymph node	23	46	12	42.9	35	44.9
Multiple sites	10	20	5	17.9	15	19.2
Pleural fluid	7	14	2	7.1	9	11.5
Brain	2	4	6	21.4	8	10.3
Bone	3	6	2	7.1	5	6.4
Spinal	2	4	-	-	2	2.6
Femur	-	-	1	3.6	1	1.3
Liver	1	2	-	-	1	1.3
Lung	1	2	-	-	1	1.3
Pericardial fluid	1	2	-	-	1	1.3

**Table 5 ijerph-20-00758-t005:** EGFR mutations present in brain metastasis samples.

Exon No.	cDNA Change	Amino Acid Change	Type of Mutation	Number of Samples
19	c.2235_2249del	Glu746_Ala750	deletion	1
19	c.2236_2250del	Glu746_Ala750	deletion	1
19	c.2239_2248delinsC	Leu747_Ala750>Pro	deletion and insertion	1
21	c.2573T>G	L858R	substitution	3

## Data Availability

Not applicable.

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
