# Peer review of "Mutational Analysis of EGFR Mutations in Non-Small Cell Lung Carcinoma—An Indian Perspective of 212 Patients"

_ijerph, 2022, doi:10.3390/ijerph20010758_

Round 1
Reviewer 1 Report
The manuscript entitled "Mutational analysis of EGFR Mutations in Non-small cell lung carcinoma- An Indian Prospective of 212 patients" the work describes the EGFR mutations among lung cancer patients in the Indian community and the response to tyrosine kinase inhibitors therapy in those mutations. Overall the work is informative as a general comment the authors are advised to adhere to the International Journal of Environmental Research and Public Health author's instructions (https://www.mdpi.com/journal/ijerph/instructions)
1) The abstract is too lengthy (i.e., 434), and stick to the recommended 200 words maximum.
2) The figure lacks sufficient resolution, images should be a minimum of 1000 pixels in width/height or a resolution of 300 dpi. Figure 3 seems compressed. It is advised to either use BioRender or a good image editing tool that would not alter the final image dpi.
3) Page 2 line 79- Ethical statement for the use of human subjects is inadequate.
Since the research involves human subjects the authors are requested to write an appropriate ethical statement including the project identification code, date of approval, and name of the ethics committee or institutional review board must be stated in the section ‘Institutional Review Board Statement’ of the article.
Best
Author Response
the changes are been made and updated in the manuscript

Reviewer 2 Report
In this manuscript, the authors did a retrospective study to gain an understanding of the spectrum of EGFR mutations. This is a well written text; the design is appropriate, however there are some key elements of the research that were overlooked and should be considered to improve the quality prior its publication.
“p-value”: use italics in “p”; Unify throughout the text: Use “Figure” or “Fig”
Line 70: ¿Did you include 212 or 214 patients?
Table 1: It is not clear the way you reported the p-value. ¿Why you did not write this value for all groups?
Line 158: “Figure 2. There was 1 (1.22%) sample which had 3 mutations and 6 (7.32%) samples which
had 2 mutations…” This information does not coincide with the graph.
Table 3 and figure 3. Please use the same number of significant figures in the table, the graph and the text (5.9 or 6; 14.28 or 14…).
Figure 3 is not referenced into the text. This must be included with the explanation.
¿What is the possible explanation of your findings 1 where the highest prevalence of EGFR mutations was seen in males over 60 years of age (67.5%)?
Considering these studies could have a significant impact in medicine, it would be extremely interesting to better explain them in the discussion, trying to put emphasis in the importance of these results regarding the differences about age, gender... Besides, it would be really interesting to deep in the causes of these mutations in certain groups (even if the theories are just hypothesis).
¿Did you have any inclusion/exclusion criteria for selecting the patients?
Author Response
The changes were made and updated in the manuscript

Reviewer 3 Report
Amrit Kaur Kaler and coworkers performed a mutational analysis of EGFR gene in biopsy samples of non-small cell lung carcinoma (NSCLC) patients. The case series is large (212 patients) and the mutation screening resulted in 38.67% of specimens carrying EGFR mutations. The correlation among the mutational status, demographic and histological characteristics as well as the localization of metastases in patients with EGFR mutations was evaluated and discussed. However, as this is a retrospective descriptive study, it is necessary to discuss if and how the EGFR mutational status was correlated with overall survival and clinical response in patients treated with tyrosine kinase inhibitors (responders versus non-responders). Reference 14 in the discussion refers to the results obtained in a work previously carried out by other authors.
Moreover, there are some typos throughout the text. For example: number of case (214 in line 70); number of older adults ˃60 (in line 145); reported percentages of EGFR mutation rate (lines256-258).
Author Response

(The authors gave the same response as above.)

Round 2
Reviewer 3 Report
Amrit Kaur Kaler and coworkers provided a revised version of the manuscript according to some of my comments. The manuscript is well written and, as the authors state, this study expands our knowledge of the spectrum of EGFR mutations among NSCLC patients in the Indian scenario. In my opinion, the manuscript may be published considering that the correlation with EGFR mutational status and treatment response would be very interesting to translate these results into clinical practice.